# Out-of-Distribution Detection using Vision Transformers

**Xuan Li**[1]                                                    XUAN.LI2@MAIL.MCGILL.CA

[1] *School of Computer Science, McGill University, 3480 Rue University, Montreal, Quebec H3A2A7, Canada*

**Christian Desrosiers**[2]                              CHRISTIAN.DESROSIERS@ETSMTL.CA

[2] *Department of Software and IT Engineering, Ecole De Technologie Superieure, 1100 Notre-Dame Street West, Montreal, Quebec H3C1K3, Canada*

**Xue Liu**[1]                                                        XUELIU@CS.MCGILL.CA

**Editors:** Under Review for MIDL 2021

## Abstract

Vision-based transformers have achieved comparable results to CNN models in tasks including object detection, image classification, and semantic segmentation. However, their performance in detecting Out-of-Distribution (OOD) samples during inference has not been fully evaluated. OOD detection plays an important role in safety-critical applications such as medical image analysis. In this paper, we evaluate 4 transformers on 2 open-sourced medical image datasets. Our results demonstrate the insufficient OOD detection performance of the transformers. Hence, future research in improving OOD detection should be encouraged.

**Keywords:** Out-of-Distribution detection, vision transformer, anomaly detection

## 1. Introduction

Vision-based Transformers have achieved wide popularity in tasks such as image classification (Dosovitskiy et al., 2020), object detection (Carion et al., 2020), and semantic segmentation (Pinaya et al., 2021). However, their robustness against abnormal data is not studied yet. This is because many closed-world tasks have the training and testing data drawn from the same distribution. We consider such distribution *In-Distribution*. But real life data are uncontrollable, and may come from a complete different distribution. We refer such data is from *Out-of-Distribution* (OOD). OOD data can falsely cause models to generate over-confident predictions (Guo et al., 2017), which raises concerns in safety-critical applications such as autonomous driving, medical diagnosis, and security screening (Amodei et al., 2016). In this paper, we evaluate the OOD detection performance on two popular vision transformer models over four different architectures, namely Vision Transformer (Dosovitskiy et al., 2020), Data-efficient image Transformer (Touvron et al., 2020) with multi-head, soft-distillation, and hard-distillation, on two well-known open-sourced skin lesion datasets HAM10000 (Tschandl et al., 2018) and DermNet (Oakley, 2016). Our experiments reveal that even though transformers have achieved impressive results in closed-world tasks, their OOD detection is still insufficient to be deployed in safety-critical applications. Future research into this field should be encouraged.

## 2. Problem Statement

**Pre-training** Given a training image $x \in X$ and a label $y \in Y = \{1, ..., K\}$ follows the data distribution $P_{in}(x, y)$ (as in-distribution). We pre-train a transformer $f_{\theta}(\cdot)$ using cross-entropy loss, where $\theta$ denotes the model parameter. After training finishes, $\theta$ will be fixed for the rest of the experiments. Note, OOD data is kept unavailable during pre-training.

**OOD Detection** We consider a disjoint dataset which follows a different distribution $P_{ood}(x)$. We sample images from a mixture distribution $Q(x, z), z \in \{0, 1\}$, where $Q(x, z = 1) = P_{in}$ and $Q(x, z = 0) = P_{ood}$, respectively. We seek the answer: Given an image $x$ drawn from the mixture distribution $Q(x, z)$, can the transformer distinguish if the image is from in-distribution or OOD based on the maximum predicted softmax probabilities.

## 3. Experiments

**Data** two different datasets HAM10000 (Tschandl et al., 2018) and DermNet (Oakley, 2016) are evaluated. The HAM10000 contains 25,332 skin lesion images taken from dermoscopes from 8 lesion classes. We treat images of 1 class Melanoma as OOD due to its severity and rareness, and the rest 7 classes as in-distribution. The DermNet contains 22,494 images taken from standard cameras from 23 lesion classes. We treat 4 classes with less than 500 images each as one single OOD and the rest 19 classes as in-distribution. A 90%-10% training-testing split is made on the in-distribution data, and the testing set is used as the in-distribution set in *OOD Detection*.

**Transformers** We compare models from Vision Transformer (ViT) (Dosovitskiy et al., 2020) and Data-efficient Image Transformers (DeiT) (Touvron et al., 2020). Specifically, for DeiT, we choose models including the original dual-head $DeiT_{orig}$, soft distillation $DeiT_{soft}$ where output is trained to match the teacher model distribution, and hard distillation $DeiT_{hard}$ where output is trained with the *argmax* of teacher model output. ResNet34 (He et al., 2016) is pre-trained as the teacher model. All codes including hyper-parameter setups are available online [1].

**Evaluation** We use metrics $AUROC$ area under the ROC curve; $AUPR_{in}$ precision recall curve where in-distribution is positive; $AUPR_{ood}$ where ood is positive; and $FPR95$ false positive rate when true positive rate is as high as 95%.

## 4. Results

The results are presented in Table 1. Overall transformers have inadequate performance when detecting OOD data. This might be due to the fact that transformers generally have more parameters than CNNs need to be trained. It is therefore requiring more data to train transformers as well. Moreover, unlike natural images, medical images such as skin lesions differ primarily in low-level features such as biological features and contours. Flattened features in transformers may inevitably lose such information. Therefore, the OOD detection performance for transformers is only comparable with ResNet34, but lower than ResNet152.

---

1. Code for reproducibility https://github.com/Shaunlipy/vision-ood

Table 1: Results for OOD detection

| Method | HAM10000 | | | | DermNet | | | |
|---|---|---|---|---|---|---|---|---|
| | $AUROC$ | $AUPR_{in}$ | $AUPR_{ood}$ | $FPR95$ | $AUROC$ | $AUPR_{in}$ | $AUPR_{ood}$ | $FPR95$ |
| ViT | 57.28 | 28.8 | 79.77 | 91.85 | 53.32 | 41.30 | 63.93 | 93.49 |
| $DeiT_{orig}$ | 57.92 | 30.24 | 79.76 | 92.83 | 54.29 | 42.06 | 64.59 | 93.58 |
| $DeiT_{soft}$ | 58.28 | 32.01 | 79.39 | 94.04 | 53.13 | 41.27 | 63.92 | 93.72 |
| $DeiT_{hard}$ | 58.20 | 29.84 | 80.18 | 91.36 | 53.63 | 41.48 | 64.27 | 93.63 |
| ResNet34 | 59.48 | 39.69 | 81.18 | 91.07 | 53.48 | 42.61 | 63.70 | 93.96 |
| ResNet152 | 61.87 | 48.94 | 83.13 | 89.27 | 55.73 | 44.52 | 66.03 | 92.08 |

## 5. Conclusion

We evaluate the OOD detection performance on four vision-based transformers. Resuts show insufficient performance which calls for future research.

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
