# OpenReview forum: "Out-of-Distribution Detection using Vision Transformers"
_MIDL.io/2021/Conference/Short — Submitted to MIDL 2021_

### Official Review · Reviewer_KUkS · 2021-04-23

**Confidence:** 4
**Final Rating:** 2

**Summary:**

The authors investigate the ability of transformers to detect out-of-distribution data (OOD), which is an important subject in computer vision, and still an active field of research.
4 transformers and 2 open source datasets are used, and a comparison with a Resnet baseline is conducted. It shows that transformers have an overall weaker performance when it comes to detecting OOD data.
Code has been made publically available.

**Strengths:**

I found the paper interesting on several aspects:
- First, the question of detecting OOD is still an active field of research, and although some methods have been described and evaluated with classical CNN, transformers are quite new (with the main 3 references in the paper dating back to 2020 or 2021) and questions like calibration of such transformers will inevitably be raised.
- Transformers are also quite new to the community, and although the authors do not present positive results on the ability of transformers to detect OOD, it could lead to fruitful discussions for further research on the subject.
- Experiments have been conducted on 2 different open source datasets.
- Code has been made available


**Weaknesses:**

- No new methodological development is presented in this paper.
- It remains a bit unclear how the output of the transformer was actually used to detect OOD. I assume that a threshold was applied to the raw output of the network, to decide whether a sample was part of the OOD set or the in-distribution set, which is the easiest way to tackle the problem.
- Therefore, I would have liked to see one other method for OOD detection presented, e.g., ensembling several models, adding test-time dropout, etc. I think this would strengthen the paper. I think that it does not represent too much work, as ensembling several models for instance requires training several models, but is then easy to implement for OOD detection.
- Some background should be added about OOD detection, even though I understand it is hard to cover everything given the page limit.

**Deanonymize Review:**

no

**Detailed Comments:**

On top of bullets 2, 3 and 4 of the weaknesses section that should be adressed, some minor typos should be fixed.

It looks like the wrong link to the datasets URL has been given during submission.

**Justification Of The Rating:**

For now, I give the paper a weak reject,
As no new methodological development is presented, I think there should be more than one OOD detection method implemented.
However, if the authors manage to adress the points I raised, I think the paper could be an interesting contribution to the conference.

**Paper Type:**

validation/application paper

**Special Issue:**

no

---

### Official Review · Reviewer_y3Rx · 2021-04-28

**Confidence:** 3
**Final Rating:** 3

**Summary:**

The paper investigates the (near) out-of-distribution detection performance of different vision transformers compared to regular CNNs on medical datasets. The paper finds that vision transformers generally have worse OOD detection performance to their CNN counterparts and reasons that this might be related to the increased data needs of ViT-like models.

**Strengths:**

The paper investigates important AI-safety questions of whether new vision transformer based models are capable of detecting near OOD samples. The paper makes good comparisons to reasonable CNN baselines and considers multiple transformer models.

**Weaknesses:**

The conclusions of the paper seem very preliminary and are hard to judge without information about the respective models predictive performance on the classification task. Further, it would have been interesting to consider CNN-Transformer hybrid models or transformers that have been pre-trained on bigger computer vision datasets to encode relevant biases about imaging data. Lastly, the authors could consider adding an experiment to test far-OOD detection with non-medical imaging datasets.

The paper should cite [1] when introducing their OOD detection baseline.

[1] Hendrycks, Dan, and Kevin Gimpel. "A baseline for detecting misclassified and out-of-distribution examples in neural networks." arXiv preprint arXiv:1610.02136 (2016).

**Deanonymize Review:**

no

**Justification Of The Rating:**

The paper considers relevant AI-safety questions around the capability of detecting near-OOD samples in the light of novel transformer-based vision architectures. Even though the work is preliminary it shows promising discussion grounds for the MIDL community.

**Paper Type:**

validation/application paper

**Special Issue:**

no

---

### Meta-Review · Area_Chair_Ux33 · 2021-05-10

**Recommendation:** Reject
**Confidence:** 4

**Metareview:**

It appears as if both reviewers found some positive aspects and adapting and evaluating known techniques for OOD detection with either transformers or standard CNNs. The results are considered to be not exactly exciting and more work would be required to come to a more insightful conclusion. In summary I believe the paper is not yet quite ready for publication and could be further improved for a future date.

---

### Decision · Program_Chairs · 2021-05-11

Reject